Feature extraction algorithm of an irregular small celestial body in a weak light environment

Cao Menglong
Gao Yue 1272173369@qq.com
School of Automation and Electronic Engineering, Qingdao University of Science and Technology , Qingdao, Shandong , China
Ashraf Imran
Electronic publication date: 2023 Jan 18
Publication date: 2023
Volume: 9
Electronic Location ID: e1198
Received 2022 May 30; Accepted 2022 Dec 5
Copyright: © 2023 Cao and Gao
Copyright year: 2023
Copyright holder: Cao and Gao
License: This is an open access article distributed under the terms of the Creative Commons Attribution License, which permits unrestricted use, distribution, reproduction and adaptation in any medium and for any purpose provided that it is properly attributed. For attribution, the original author(s), title, publication source (PeerJ Computer Science) and either DOI or URL of the article must be cited.
License URL: https://creativecommons.org/licenses/by/4.0/

Keywords: Feature extraction, Celestial image, ORB feature point detection, Bilateral filtering

Funding: Key Scientific Issues of Transformative Technologies Theory and Method of Intelligent Attachment of Non-cooperative Targets in Space Intelligent Attachment of Non-cooperative Targets 2019YFA0706502-3 This work was supported by the National Key R&D Program “Key Scientific Issues of Transformative Technologies”, the “Theory and Method of Intelligent Attachment of Non-cooperative Targets in Space”, and the project of Intelligent Attachment of Non-cooperative Targets (No. 2019YFA0706502-3). The funders had no role in study design, data collection and analysis, decision to publish, or preparation of the manuscript.

==============================
This study focuses on creating a crater-matching algorithm to improve the matching rate and address the phenomenon of insufficient feature extraction and mismatching of irregular celestial objects and crater edge information on the dim surface of celestial bodies images. These images were captured by the detector’s navigation camera. In order to improve the brightness and clarity of the images, the target images were filtered, denoised, and image-enhanced using the bilateral filtering method and improved histogram equalization algorithm, successively. Then, the enhanced image was extracted and matched using the ORB feature point detection algorithm based on scale invariance, and the feature point mismatch was processed by the Hamming distance screening method. The simulation results revealed that the optimization algorithm effectively improved the imaging quality of the target image in dark and weak light environments, increased the number of feature points extracted, reduced the mismatch of effective feature point pairs, and improved the matching rate.

Introduction

The detection of small celestial bodies is significant for space resource utilization and helps to reveal the origin and evolution of the solar system (Zhang, Xu & Ding, 2021; Kohout et al. (2018)). Small celestial bodies have smaller masses, weak gravitational forces, and are easy to escape, so collecting resources from small celestial bodies is simpler than from the moon (Cui, Jia & Zhu, 2021). Future small-body exploration missions will require more precise landing capabilities, including sample collection, return missions and landings at scientifically feasible locations. However, the precise landing of small celestial detectors is an enormous technical challenge (Shao et al., 2016).

Autonomous optical navigation can reduce the complexity of operations, reduce mission costs, and improve the efficiency of small celestial object detection tasks. It has developed into the main navigation method for deep space exploration (Cui, Zhu & Cui, 2009). In the past, many small celestial detection missions performed autonomous navigation using optical feature information at different stages, which proved the feasibility of using image features to realize autonomous optical navigation (Christian, 2015). According to our future development plan for deep space exploration, it is of great theoretical and practical significance to carry out the identification and extraction of optical navigation features of small celestial bodies. During the probe’s landing stage, pits and rocks on the surface of small celestial bodies are the main optical navigation features. During the landing, the image captured by the optical camera depends on the illumination conditions, imaging angle, and imaging time. Image quality is generally lower with poorer conditions, making it extremely difficult to extract the feature of the target image (Cui et al., 2020). Moreover, the surface features of small celestial bodies are complex, and include craters, rocks, slopes, ravines, and other surface features. It is very difficult to identify and match a variety of feature information simultaneously, and it is difficult to ensure extraction accuracy and real-time performance (Yu, Cui & Tian, 2014). The craters on the surface of small celestial bodies are annular craters formed by the impact of meteorites on the surface of celestial bodies. They are the basic source of information concerning the geological and surface characteristics of small celestial bodies (Marques & Pina, 2014). Therefore, this article focuses on the feature extraction of annular pits on the surface of small celestial bodies. This requires the development of image processing algorithms for dim targets that can accurately extract the boundary information of craters on the surface of celestial bodies and obtain accurate navigation information.

The key technologies for the feature extraction of craters on the surface of celestial bodies include edge detection, gradient filtering, and neural network algorithms (Cui et al., 2018). Xie et al. (2022) proposed the idea of combining adaptive histogram equalization and an ORB algorithm to obtain better feature point quality and matching efficiency under dynamic scene moving target recognition. Tian & Yu (2017) and Tian, Li & Song (2018) proposed a new terrain-relative navigation (TRN) method and a new crater-to-3D crater model matching method for asteroid fixed-point landing missions. In this method, the distance and direction between features are used as elements to form a 3D feature description set, which improves the matching rate of terrain features under the condition of the sparse 3D point cloud. Shao et al. (2020) proposed a crater matching algorithm based on feature descriptors, which is suitable for navigation during descent. Luo et al. (2019) introduced the process of the ORB algorithm, which has more stable feature extraction and stronger anti-interference ability.

This article focuses developing feature extraction methods based on artificial intelligence and machine learning to better capture the effective features of irregular and weak targets with consideration for their unique imaging characteristics. In order to address the problems of rotation, scaling, and deformation between matched image pairs, it was necessary to carry out fault-tolerant processing analysis, study the invariant properties of the orthographic projection of target features, and reduce feature mismatches (Cheng et al., 2013; Zhang & Koch, 2013). The processed image was extracted and matched using the ORB feature point detection algorithm based on scale invariance by denoising and enhancing the original target image. The Hamming distance screening method was used for feature point mismatch processing, reducing feature mismatch, segmenting effective features from images with limited information, improving the matching rate, and extracting more accurate navigation information.

Materials and Methods

Target feature recognition image preprocessing technology

The deep space environment is complex making noise filtering an important step in image preprocessing (Zhang et al., 2019). The image noise of small celestial bodies is high-frequency, so low-pass filtering is used to suppress the noise. The edges of the surface pits captured by the navigation camera are important observation points. However, edge area information of deep sky images is missing in dark and weak light environments. Therefore, a smoothing algorithm with edge-preserving performance was selected to denoise the target image. Bilateral filtering is a kind of nonlinear filtering (Liu & Pang, 2021; Wang et al., 2021). It uses the weighted-average method and the weighted average of the surrounding pixel brightness values to represent the intensity of a pixel. The weighted average was based on a Gaussian distribution. Most importantly, the weight of bilateral filtering considers the Euclidean distance of the pixel as well as the radiation difference in the pixel range domain. This can achieve the effect of maintaining edges and reducing noise and smoothness. Therefore, bilateral filtering was used to denoise the target celestial image.

Image denoising technology based on the bilateral filtering method

Bilateral filtering is a nonlinear smoothing filter obtained according to the spatial distribution of image pixels and the similarity between pixels, which can denoise the image and achieve edge protection. In the bilateral filter, the output pixel value after denoising of the central pixel is obtained by combining the weighted values of its neighboring pixel values. See Eq. (1):

(1) g(x,y)=∑k,lf(k,l)ω(i,j,k,l)∑k,lω(i,j,k,l)

where f(k,1) is the neighborhood pixel value; g(x,y) is the output pixel value; ω(i,j,k,l) is the weighting coefficient of neighborhood pixel (k,l) with (i,j) as the center, which is determined by the product of definition domain core and value domain core.

The definitions of domain core d(i,j,k,l) and value domain core r(i,j,k,l) are shown in Eqs. (2) and (3):

(2) d(i,j,k,l)=exp[−(i−k)2+(j−l)22σd2]

(3) r(i,j,k,l)=exp[−‖f(i,j)−f(k,l)‖22σr2]

where σd, σr is the filtering radius.

σd and σr are multiplied to obtain the bilateral filtering weight function depending on the data, as shown in Eq. (4):

(4) ω(i,j,k,l)=exp[−(i−k)2+(j−l)22σd2−‖f(i,j)−f(k,l)‖22σr2]

Figure 1 shows the comparison of celestial surface images before and after bilateral filtering. The information of a large number of chaotic small pits on the surface of the celestial body in the figure is smooth and blurred, and the problem of uneven illumination has also been improved to a certain extent after the original image is denoised by bilateral filtering. Detailed information at the edge of the pit with significant information is still relatively apparent, which achieves the purpose of edge preservation and smoothness. It is beneficial to further extract information from the image. However, because the overall brightness of the image is low, the image should be enhanced.

Figure 1 Comparison of celestial images before-and-after filtering.

Figure source credit: Copyright ESA 2010 MPS for OSIRIS Team MPS/UPD/LAM/IAA/RSSD/INTA/UPM/DASP/IDA.

Image enhancement technology based on improved histogram equalization algorithm

The primary purposes of image enhancement is to improve the quality and recognition of the image, highlight the parts or features of interest in the image, suppress secondary or unserviceable information in the image, and make the image easy to observe and further analyze and process (Li et al., 2021). To address the problem of uneven light and low overall brightness of target celestial images captured by navigation cameras, we propose an improved histogram equalization image enhancement algorithm (HEEF) based on edge information fusion (Dong, Ding & Xu, 2018). The processing results of the Laplace algorithm (Uoosefian et al., 2020) and the processing results of the classical histogram equalization (HE) algorithm (Jebadass & Balasubramaniam, 2022) are proportionally weighted and fused. The algorithm is simple to implement but effective for image detail enhancement. Histogram equalization is a common image enhancement method, which adjusts the gray value of the original image so that the gray histogram of the image is transformed from a non-uniform distribution to a uniform distribution in the gray range to improve the image area. The brightness and contrast enhancement is simple and easy to achieve.

As shown in Fig. 2B, although the classical histogram equalization algorithm improves the image quality to a certain extent, it still leads to the lack of some information, and the imaging quality does not meet the actual requirements, so it needs to be improved.

Figure 2 Before-and-after image enhancement.

Figure source credit: Copyright ESA 2010 MPS for OSIRIS Team MPS/UPD/LAM/IAA/RSSD/INTA/UPM/DASP/IDA.

Improving the histogram equalization image enhancement algorithm using edge information fusion is a histogram equalization method that introduces edge information fusion technology.

The mathematical model of the algorithm is shown in Eq. (5). The only parameter of the model is the weighting ratio, which utilizes a few parameters and is a simple calculation.

(5) IHEEF=λIHE+(1−λ)ILaplace

where λ is the weighting parameter. When λ=0, the model degenerates to the Laplace sharpening algorithm, and when λ=1, it degenerates to the classic HE algorithm. Therefore, the parameter value range is 0<λ<1, and λ=0.5 was used in this article. In the literature (Dong, Ding & Xu, 2018), three different parameters of λ=0.2, λ=0.5, and λ=0.8 were used for experimental comparison. The results show that the enhancement effect was better when λ=0.5 was used; therefore, this was selected as the optimization parameter of the algorithm in this study.

After the image was enhanced by the improved histogram equalization algorithm, the gray level probability density function value of the image was one, which satisfies the uniform distribution and enhanced image contrast. As shown in Fig. 2C, the overall brightness of the image processed by the improved histogram equalization algorithm was significantly improved, the contrast of the area was effectively enhanced, each area in the image was clearer, and the edge details were more prominent. While improving the brightness of small celestial bodies, the edge information of small celestial bodies can be well preserved, and the information of pits on the surface of celestial bodies was more significant, which can assist in the identification and extraction of pit features on the surface of celestial bodies.

Image navigation feature extraction technology

Surface feature information can be obtained using the Harris corner detection algorithm. However, the distribution of the feature points extracted by this algorithm is concentrated on the edges of rocks and craters, so it can only match feature points in the local area. Therefore, feature extraction algorithms based on scale invariance are widely used in feature recognition and matching. As the most widely used key point detection and description algorithm, SIFT (Scale-Invariant Feature Transform) has the advantages of rotation, scale, translation, viewing angle and brightness invariance, which is conducive to the effective expression of target feature information. However, SIFT also has major disadvantages, including that the alternations are difficult to achieve without hardware acceleration or a dedicated graphics processor. The SURF (Speeded Up Robust Features) operator is an improvement of SIFT. SURF simplifies some of the operations in SIFT. Although the robustness of the algorithm is increased, there is no great improvement in real-time performance. The running time of the ORB feature description algorithm is much better than that of SIFT and SURF, and it can be used for real-time feature detection. The ORB feature is invariant to scale and rotation, and it is also invariant to noise and perspective affine. Its performance makes it possible to use ORB in a wide range of application scenarios for feature description (Rublee et al., 2011). Therefore, the ORB feature description algorithm was used in this study.

ORB feature extraction combines the detection method of FAST feature points with a brief feature descriptor. This method can improve and optimize the features based on the original images. First, it uses the FAST feature point detection method to detect feature points and then uses the Harris corner measurement method to select the N feature points with the largest Harris corner response values from the FAST feature points. The response function of the Harris corner is defined as:

(6) R=detM−α(traceM)2

where detM is the determinant of matrix M=[ACCB]; traceM is the direct trace of matrix M; α is a constant with a value range of 0.04–0.06. A=∑w(x,y)Ix2; B=∑w(x,y)Iy2; C=∑w(x,y)IxIy. w(x,y) is the image window function, Ix,Iy are the changes in the horizontal and vertical directions of the feature points, respectively.

The number of effective feature points determines the matching quality between image frames. Using the ORB algorithm to extract features requires two parts: FAST key points and BRIEF (Binary Robust Independent Elementary Features) descriptors. The traditional FAST key point extraction does not have enough distinguishability and is greatly affected by the distance, however, the directional key points (Oriented FAST) can extract key points more accurately and effectively. The steps to the realization of the Oriented FAST algorithm are: (1) Select a pixel α in the image, assuming its brightness is Iα, and set a pixel threshold T ( T can be taken according to the specific situation after multiple tests).

(2) Select a circle with a radius of three pixels centered on a pixel α, which is composed of 16 pixels.

(3) If the brightness of 12 consecutive points exceeds the interval ±T (i.e., greater than Iα+T or less than Iα−T), the pixel α can be used as a feature point.

(4) In the region D with a radius of three centered on a pixel α, the moment defining the region is:

(7) mpq=∑x,y∈DxpyqI(x,y) p,q∈{0,1}.

The centroid of the image in this area can be found through the moment:

(8) B=(m10m00,m01m00).

(5) The direction vector OB connecting the image block center O and the centroid B is the direction of the feature point, which is defined as:

(9) θ=arctan⁡(m01/m10).

(6) Repeat the above steps and do the same for each pixel.

Once the feature points are extracted using the above method, the image area around the extracted feature points must be further described using the BRIEF descriptor. The BRIEF descriptor is a binary descriptor, and its description vector consists of 0 and 1. Under the premise of the above feature points with direction information, the calculated BRIEF descriptor also has good rotation invariance, and randomly selects 128 near the feature points. For pixel (p,q), it is arranged as a 128-dimensional binary number matrix according to the following rules:

(10) {1,p>q0,p<q.

Since the number of feature points in celestial images in dark and low light environments is not large, the brute-force matcher (Liao et al., 2021) is a suitable method. Note that the previous frame image is It, the extracted feature points are xtm,m=1,2,3…M, the next frame image is It+1, and the extracted feature points are xt+1n,n=1,2,3…N. The distance of the descriptor is measured using the Hamming distance (Wang & Zheng, 2021). In informatics, the number of characters in the corresponding position between two strings of equal length is called the Hamming distance. This article introduces the concept of Hamming distance into ORB algorithm, and determines the validity of feature points by judging the distance between two descriptors. The specific algorithm flowchart is shown in Fig. 3.

Figure 3 Algorithm flowchart.

Results

The algorithm experiments in this article were verified by a machine learning simulation. The Ubuntu 20.04 operating system was used with 50 GB of memory and a 2 GHz CPU; the operating environment was Openc v3.4.13. The pixel threshold T was set when extracting oriented fast key points and was experimentally verified to be 20% of pixel illumination Iα. The experimental part was divided into a feature point extraction experiment and analysis of a single image, the feature point matching experiment, and a before-and-after analysis between the two frames of images.

To reflect the advantages of the optimization algorithm proposed in this article in extracting and matching feature points, we conducted an experimental analysis of feature point extraction in a single image and experimental analysis of feature point matching in two before-and-after frames of images. The images were surface pictures of small celestial bodies taken by the navigation camera in dark and low light environments of deep space. Due to blurred light and low definition, the number of invalid feature points (that is, the feature points without accuracy and validity) increased, seriously affecting the matching quality of the two frames of the before-and-after images. As a result, sufficient navigation information could not be obtained, resulting in low optical navigation accuracy. Once the image was enhanced by the proposed research algorithm, the above problems were effectively solved.

Figure 4 shows an example of the camera angle of the detector navigation. The simulation experiment of the algorithm was conducted by taking the irregular celestial body image and the local detail map of the small celestial body. Figure 5 shows that the feature points were extracted and matched directly on the target image before image preprocessing. The number of feature points was small and there were many invalid points. When performing image matching, the quality of the feature point matching of the before-and-after frames largely depends on the number and quality of feature points extracted from a single image. The feature matching quality was not high, and we used the matching feature points as optimized by the algorithm. Therefore, it was necessary to filter and denoise the image and image enhancement.

Figure 4 Example of detector camera shooting.

Figure source credit: Copyright ESA 2010 MPS for OSIRIS Team MPS/UPD/LAM/IAA/RSSD/INTA/UPM/DASP/IDA.

Figure 5 Feature points and matching of unprocessed images.

Figure source credit: Copyright ESA 2010 MPS for OSIRIS Team MPS/UPD/LAM/IAA/RSSD/INTA/UPM/DASP/IDA.

It can be seen from Fig. 6 that the matching quality of image feature points enhanced by bilateral filtering and denoising and the improved histogram equalization algorithm was improved. The number of invalid feature points decreased as the number of feature points increased with the image enhancement resulting from the use of our algorithm, regardless of whether it is a small celestial object image or a local image of surface features, and the point pairs were effectively matched. As the effective match rate increased, the match quality was improved.

Figure 6 Feature points after algorithm enhancement.

Figure source credit: Copyright ESA 2010 MPS for OSIRIS Team MPS/UPD/LAM/IAA/RSSD/INTA/UPM/DASP/IDA.

According to the feature point extraction and matching data in Table 1, it can be seen that the number of effective feature points and the number of effective matching feature point pairs after optimization were significantly higher than those before optimization. The proportion of effective feature points increased from 69.92% to 83.28%. The proportion of effective matching feature point logarithms increased from 66.67% to 82.67%. The optimization algorithm can effectively improve the imaging quality of the target image in a dark and weak light environment, increase the number of feature point extraction, reduce false matching, and improve the matching rate.

Table 1 Feature point extraction and matching data table.

Image	Number of feature points	Number of effective feature points	Number of matching feature point pairs	Number of effective matching feature point pairs	Percentage of effective feature points (%)	Percentage of the logarithm of effective matching feature points (%)	
Before HEEF algorithm optimization	276	193	201	134	69.92%	66.67%	
After HEEF algorithm	291	212	239	173	72.85%	72.24%	
After algorithm optimization in this paper	341	284	300	248	83.28%	82.67%	

To prove the feasibility of the optimization algorithm proposed in this article, the MATLAB simulation software was used to verify that the celestial image retained the edge information integrity of the surface pits after the algorithm was optimized. Figure 7 clearly shows that the boundary information of the celestial image was clearer and richer after applying the algorithm for better optimization. The optimization algorithm proposed in this article is feasible and effective in solving the difficulty of feature point extraction. It can extract an increasing amount of accurate navigation information for optical navigation. At the same time, the celestial image preprocessing algorithm also has great practicability in the centroid extraction algorithm for irregular small celestial bodies and ultimately improves the navigation accuracy.

Figure 7 Comparative analysis of boundary information for celestial images.

Conclusions

The experimental data shows that if the feature point extraction and matching are performed directly on the image captured by the navigation camera, there are few feature points and too many invalid feature points, making feature extraction difficult and reducing the usefulness of the extracted information. In this study, the original image was enhanced using bilateral filter denoising and the improved histogram equalization algorithm. The ORB feature point detection algorithm based on scale invariance was used to extract and match features, and the Hamming distance screening method was used to address the mismatch of the feature points. Through simulation experiments, the proposed algorithm ensures the integrity of the image information. The total number of feature points and the number of valid feature points were significantly improved, increasing the feature matching rate. The proposed algorithm can provide richer and more accurate navigation information so that the detector can better adjust the pose and find more accurate landing landmarks. The algorithm also has great practicability and feasibility in extracting the centroid of irregular small celestial bodies.

Supplemental Information

Supplemental Information 1 Raw data.

Click here for additional data file.

Supplemental Information 2 Simulation code.

Click here for additional data file.

Additional Information and Declarations

Competing Interests

Author Contributions

Data Availability

The authors declare that they have no competing interests.

Menglong Cao conceived and designed the experiments, analyzed the data, prepared figures and/or tables, authored or reviewed drafts of the article, and approved the final draft.

Yue Gao conceived and designed the experiments, performed the experiments, analyzed the data, performed the computation work, prepared figures and/or tables, authored or reviewed drafts of the article, and approved the final draft.

The following information was supplied regarding data availability:

The code and raw data are available in the Supplemental Files.

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
