# Peer review of "Feature extraction algorithm of an irregular small celestial body in a weak light environment"

_PeerJ Computer Science, doi:10.7717/peerj-cs.1198_

## Round 0.1 · original submission · Major Revisions

Based on the reviewers' comments, the authors are requested to make "major" revisions.

Reviewer 1 ·

Basic reporting

The review of the literature is well drafted, so the reader have adequate background about the topic.
Abstract and Introduction is c;ear and unambiguous in nature and it is well written.

Experimental design

The methodology used in the paper is novel and the experimental model was well explained. Sufficient details were also provided about the implementation. Originality of the implementation was carried out.

Validity of the findings

Figure, table with the data are clearly given in the paper.
Conclusion with the result are well stated, linked to the article focused.

Additional comments

Overall the paper is well written and it shows the future direction for carrying further research in this filed of study.

·

Basic reporting

The English grammar needs to be modified. In addition, formula 1 and formula 4 show errors and need to be modified, which will affect the review.

Experimental design

The experimental design part is good, and efficient algorithms are presented.

Validity of the findings

The algorithm is valid.

Additional comments

The grammar, expressions, formulas, references, etc. of the full text still need to be improved. It is recommended to accept after minor revisions.

Reviewer 3 ·

Basic reporting

- The authors would need to check for grammatical and continuity issues. e.g.: in lines 46-47: the authors should clarify/re-write this sentence.
- Line 42: the authors might consider changing “my country's future” to either “our” or “China’s”
- Sufficient field background are not provided. There is a lot of literature on SIFT and ORB algorithms for edge detection. e.g., in lines 64-66: the authors might want to consider re-phrasing the findings of [11]
- Mathematical equation typesetting is hard to read: e.g., Eq (1), (2) are hard to read. Perhaps the authors can typeset on some equation editor?
- Some issues with self-containment: the authors might want to add some more background information on ORB. Also, eg., it is not clear what A, B, C and D in the matrix M of Eq (6) refer to.
- Definition and theorems: the authors might want to add a couple of lines explaining the intuition for the domain cores. Also, it will be helpful to explain why omega (Eq. 4) is a valid weighting function, i.e., does it lie between 0 and 1? In Eq (5), is there a specific reason for picking lambda = 0.5?

Experimental design

- Research question is meaningful and relevant.
- However, I think there are other papers which have similar ideas in them. As an example, the paper "Overview of Image Matching Based on ORB Algorithm" with DOI: 10.1088/1742-6596/1237/3/032020 has a lot of the technique in common. It is advised that the authors specify what gap the present paper fills.

Validity of the findings

- The analysis is based on well-founded literature on the ORB algorithm. However it is unclear how the Hamming distance method is utilized - the authors have mentioned it in the abstract and the summary.

Additional comments

Overall, the authors might need to work on rewriting some sections where they have reviewed other work. Adding in some more references is recommended.

Reviewer 4 ·

Basic reporting

1# The authors of this paper propose to enhance and denoise the original target image by Improved Histogram Equalization Algorithm.

2# First of all, the manuscript is quite clear and easy to understand What the authors try to do.

Experimental design

2#major In the experimental evaluation you did not consider other algorithms. You should include the results also for at least one of those models.
For example, the authors claim that they have improved the Histogram Equalization Algorithm. They should compare Improved Histogram Equalization Algorithm and Histogram Equalization Algorithm.
(Table 1 feature point extraction and matching data table)


3#major. The sample space is minimal and it is very difficult to generalize by looking at only one picture. They need to explain the size of the dataset. and test their improved Histogram Equalization Algorithm
on the large size of the irregular small celestial body image dataset.

Validity of the findings

4#major. Adding at least another couple of image data would extend the validity of the results.

Additional comments

5#minor. The flow chart and the design of the article should be done more regularly.
(Fig. 3 Algorithm flowchart)

---

## Round 0.2 · Minor Revisions

Based on the comments given by the reviewer, The authors are advised to make "Minor Revisions" and resubmit.

Reviewer 3 ·

Basic reporting

- The authors have addressed most of the concerns I had with the previous draft.
- The quality of English is much better than the previous draft, and the present manuscript has a clear narrative.
- The references make it easy to follow the arguments and provides substantial background.
- Mathematical equations have been properly typeset, which was a concern for me previously.

Experimental design

- The research question, as stated previously, is meaningful and relevant.
- The experimental design is founded on solid prior research and addresses clearly the gaps that are not satisfied by other work. I congratulate the authors for this present manuscript.

Validity of the findings

- The analysis is based on well-founded literature on the ORB algorithm. The authors still have not included any discussion on the Hamming distance method – although this has been expressly mentioned in the abstract and in the summary. I had urged the authors to add some discussion around it.
- The arguments are solid, and I think otherwise that the research is well-founded.

Additional comments

- Overall, the authors have satisfied most of my concerns, and I think this is in proper shape for publication as it stands.

Reviewer 4 ·

Basic reporting

The authors of this paper propose to enhance and denoise the original target image by Improved the Histogram Equalization Algorithm.

Experimental design

the manuscript is quite clear and easy to understand.

Validity of the findings

The idea of improving the brightness and clarity of the images is interesting. The reported results show that the proposed bilateral filtering method improved histogram equalization algorithm is more accurate than before optimizing the algorithm

Additional comments

The flow chart and the design of the article should be done more regularly.
Improve the quality of Figure 3 Algorithm flowchart



The table 1 image column should be written more clearly.
ex :
1. row = Before HEEF Algorithm Optimization
2. row = After HEEF Algorithm
3. row = After HEEF Algorithm Optimization

---

## Round 0.3 · Minor Revisions

The reviewers recommend the acceptance of your manuscript. For final draft, try to resolve minor issues from Reviewer 4.

Reviewer 4 ·

Basic reporting

1# The authors of this paper propose to enhance and denoise the original target image by Improved the Histogram Equalization Algorithm.

the manuscript is quite clear and easy to understand What the authors try to do.

Experimental design

In the experimental evaluation, you did not consider other algorithms. You should include the results also for at least one of those models.

For example, the authors claim that they have improved the Histogram Equalization Algorithm. They should compare Improved Histogram Equalization Algorithm and Histogram Equalization Algorithm

Validity of the findings

in conclusion, this model provides a new point of view on irregular celestial objects.

Additional comments

Thank you Authors for their contributions

---

## Round 0.4 · accepted · Accept

The authors have resolved the comments adequately. The paper is accepted.